## Replication

psychology

emotional expression, face processing, individual differences

**Author for correspondence:**
Benedict C. Jones
e-mail: benedict.jones@strath.ac.uk

# Are affective factors related to individual differences in facial expression recognition?

Sarah A. H. Alharbi[1,2], Katherine Button[3],
Lingshan Zhang[1], Kieran J. O'Shea[1], Vanessa Fasolt[1],
Anthony J. Lee[4], Lisa M. DeBruine[1]
and Benedict C. Jones[4,5]

[1]Institute of Neuroscience & Psychology, University of Glasgow, Scotland, UK
[2]Department of Psychology, Taibah University, Medina, Saudi Arabia
[3]Department of Psychology, University of Bath, England, UK
[4]Division of Psychology, University of Stirling, Scotland, UK
[5]School of Psychological Sciences and Health, University of Strathclyde, Scotland, UK

AJL, 0000-0001-8288-3393; LMD, 0000-0002-7523-5539;
BCJ, 0000-0001-7777-0220

Evidence that affective factors (e.g. anxiety, depression, affect) are significantly related to individual differences in emotion recognition is mixed. Palermo *et al.* (Palermo *et al.* 2018 *J. Exp. Psychol. Hum. Percept. Perform.* **44**, 503–517) reported that individuals who scored lower in anxiety performed significantly better on two measures of facial-expression recognition (emotion-matching and emotion-labelling tasks), but not a third measure (the multimodal emotion recognition test). By contrast, facial-expression recognition was not significantly correlated with measures of depression, positive or negative affect, empathy, or autistic-like traits. Because the range of affective factors considered in this study and its use of multiple expression-recognition tasks mean that it is a relatively comprehensive investigation of the role of affective factors in facial expression recognition, we carried out a direct replication. In common with Palermo *et al.* (Palermo *et al.* 2018 *J. Exp. Psychol. Hum. Percept. Perform.* **44**, 503–517), scores on the DASS anxiety subscale negatively predicted performance on the emotion recognition tasks across multiple analyses, although these correlations were only consistently significant for performance on the emotion-labelling task. However, and by contrast with Palermo *et al.* (Palermo *et al.* 2018 *J. Exp. Psychol. Hum. Percept. Perform.* **44**, 503–517), other affective factors (e.g. those related to empathy) often also significantly predicted emotion-recognition performance. Collectively, these results support the proposal that affective factors predict individual differences in emotion recognition,

but that these correlations are not necessarily specific to measures of general anxiety, such as the DASS anxiety subscale.

## 1. Introduction

Facial expression recognition plays an important role in social interaction. Although it is widely acknowledged that substantial individual differences in facial expression recognition exist, the factors that underpin these individual differences are poorly understood [1]. Many studies that have investigated this issue have focused on the role of affective factors, such as anxiety, depression, mood and empathy.

Evidence from studies investigating the relationship between affective factors and facial expression recognition has been mixed. For example, while studies of clinical samples have found that anxious or depressed people show impaired facial expression recognition (e.g. [2,3]), some studies of non-clinical samples have not observed significant correlations between facial expression recognition and anxiety or depression (e.g. [4]). Similarly, while some studies have reported that people who score higher on measures of empathy or lower on measures of autistic-like traits perform better on facial expression recognition tasks (e.g. [5,6]), other studies have not replicated these results (e.g. [1]).

Interpreting the mixed results for affective factors and facial expression recognition described above is complicated because different studies have investigated different affective factors and/or used different methods to assess facial expression recognition. Direct replications (i.e. studies using the same measures as the original work) are one way to address this difficulty because they allow for more direct comparison of results across studies [7].

In the light of the above, we directly replicated one recent study of the possible link between affective factors and facial expression recognition [1]. We chose this particular study to replicate because it considered a relatively broad range of affective factors (various measures of anxiety, depression, mood and empathy) and showed consistent results across two recently developed comprehensive facial expression recognition tasks (the emotion-matching and emotion-labelling tasks developed and described in [8]). We also chose Palermo *et al.* [1] for our direct replication because, despite these methodological strengths, the significant relationships between affective factors and facial expression recognition would not have been significant if corrected for multiple comparisons. This pattern of results suggests that the correlations between anxiety and facial expression recognition may not necessarily be robust.

Palermo *et al.* [1] reported that participants' ($N = 63$) scores on the anxiety scale of the depression anxiety and stress scales (DASS) were negatively correlated with their performance on Palermo *et al.*'s [8] emotion-matching ($r = -0.287$, $p = 0.023$) and emotion-labelling ($r = -0.255$, $p = 0.044$) tasks. By contrast with their results for anxiety, participants' performance on neither of these emotion-recognition tasks was significantly correlated with their scores on questionnaires measuring a range of other affective factors (empathy, depression, or mood). Performance on a third emotion recognition test ([9] multimodal emotion recognition test) was not significantly correlated with any of the affective factors. Based on these results, Palermo *et al.* [1] concluded that anxiety is the critical affective factor for individual differences in facial expression recognition.

Following Palermo *et al.*'s [1] results, we tested four specific hypotheses:

**Hypothesis 1.** Scores on the anxiety scale of the DASS will be significantly negatively correlated with performance on the emotion-matching task.

**Hypothesis 2.** Scores on the anxiety scale of the DASS will be significantly negatively correlated with performance on the emotion-labelling task.

**Hypothesis 3**. Performance on neither the emotion-matching nor emotion-labelling tasks will be significantly correlated with scores on the depression scale of the DASS, the positive affect scale of the positive and negative affect schedule (PANAS), the negative affect scale of the PANAS, scores on the autism quotient (AQ), scores on the empathy quotient (EQ), scores on the affective component of the basic empathy scale (BES), or scores on the cognitive component of the BES.

**Hypothesis 4.** Performance on Bänziger *et al.*'s [9] multimodal emotion recognition test will not be significantly correlated with scores on any of the affective factors.

This article received in-principle acceptance (IPA) on 26 April 2019. Following IPA, the accepted Stage 1 version of the manuscript was preregistered on the OSF at https://psyarxiv.com/fg8yz/. This preregistration was performed prior to data collection and analysis.

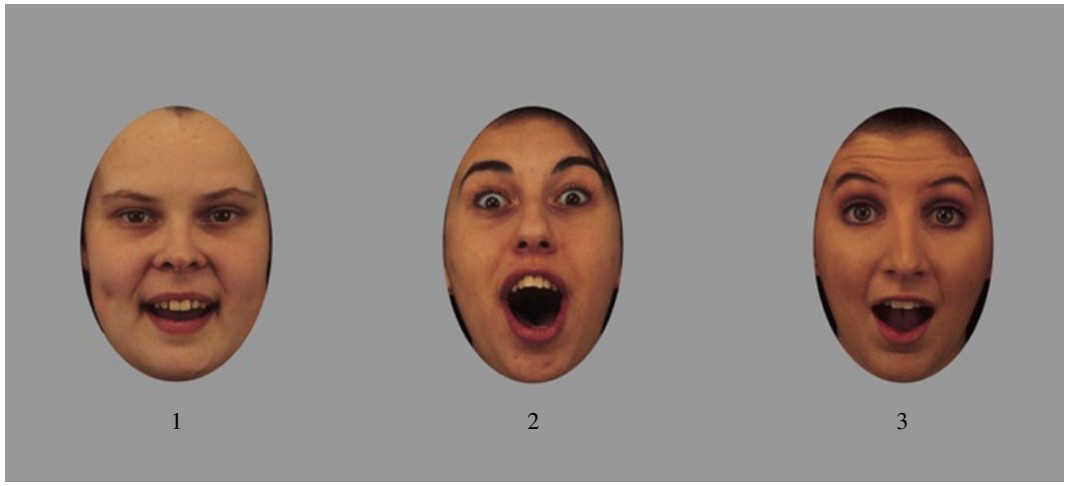

**Figure 1.** An example trial from the emotion-matching task. Participants are instructed to indicate which face is showing a different emotional expression to the other two faces (i.e. which emotion is the 'odd-one-out'). The correct answer on the trial shown is face 1.

# 2. Methods

## 2.1. Participants and justification of sample size

Palermo et al. [1] tested 63 university students (55% women, 45% men). Simonsohn [10] recommends that sample sizes in replication studies be approximately two and a half times the sample size of the original study. Consequently, we aimed to test 160 University of Glasgow students between the ages of 16 and 45 years with a similar sex ratio to Palermo et al.'s [1] original study. 168 participants in total took part in the study.

## 2.2. Emotion-matching task

We used the same emotion-matching task as Palermo et al. [1]. This is the 100-item emotional masking task originally developed by Palermo et al. [8]. Images of three different individuals (matched for sex) are presented on each trial. Two of the images (the distractor images) are shown with the same emotional expression (e.g. anger). The other image (the target image) is shown with a different emotional expression (e.g. disgust), making it the 'odd one out'. Target and distractor emotions are paired to be maximally confusable accordingly to previously published data [11]. Participants use numbered keys to indicate whether face 1, face 2, or face 3 is displaying the 'odd-one-out' emotion. Participants can respond either while the faces are presented onscreen (4500 ms) or any time up to 7000 ms time after the faces are no longer presented onscreen. The 100 trials are presented in the same order for each participant and are preceded by eight practice trials. The stimuli, target-distractor pairings, and trial order we used are identical to those in Palermo et al. [8] and described in their supplemental materials. Performance on this task is indicated by the percentage of trials on which a participant correctly identifies the target face. An example trial from the emotion-matching task is shown in figure 1. We have obtained the code and stimuli for this task from the corresponding author of Palermo et al. [1], allowing us to precisely replicate this task. Stimuli are from the Karolinska Directed Emotional Faces image database [12], and were shown in color on an iMAC12.1 at 1686 × 762 pixels.

## 2.3. Emotion-labelling task

We used the same emotion-matching task as Palermo et al. [1]. This is the 100-item emotion-labelling task originally developed by Palermo et al. [8], but with two modifications (presentation time for each face reduced from 1000 ms to 400 ms and the number of facial expressions in each emotion category being the same). Both of these modifications to the task described in Palermo et al. [8] were also made in Palermo et al. [1]. Each face is individually presented on screen. Participants use a computer mouse to select the appropriate emotion label from a set of six labels presented underneath the face (anger,

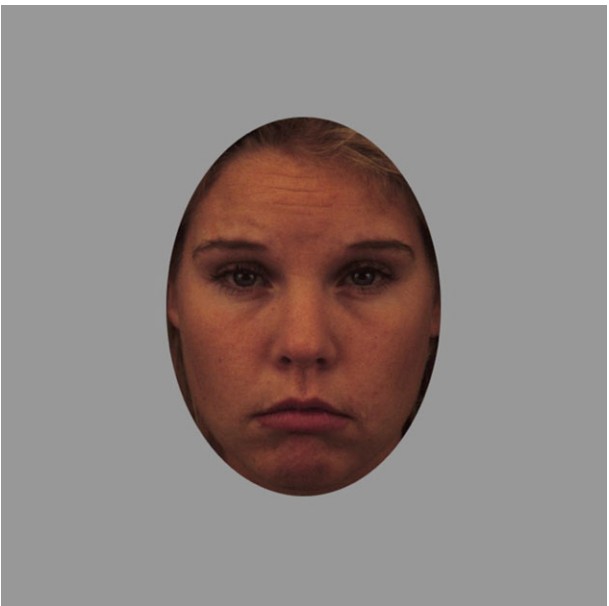

**Figure 2.** An example trial from the emotion-labelling task. Participants are instructed to select the appropriate emotion label from a set of six labels presented underneath the face (anger, disgust, fear, surprise, sadness, happiness). The correct answer on the trial shown is sadness.

disgust, fear, surprise, sadness, happiness). Responses can be made while the face is presented (400 ms) or up to 7000 ms after the face is no longer presented onscreen. The stimuli and trial order we used are identical to those used in Palermo et al. [1] and described in an email provided by the corresponding author of the Palermo et al. [1] paper. Performance on this task is indicated by the percentage of trials on which the participant correctly labels the facial expression. An example trial from the emotion-labelling task is shown in figure 2. We have obtained the code and stimuli for this task from the corresponding author of Palermo et al. [1], allowing us to precisely replicate this task. Stimuli from the Karolinska Directed Emotional Faces image database [12] were shown in color on a iMAC12.1 monitor, at $737 \times 737$ pixels.

## 2.4. Multimodal emotion recognition test

This test, described in full in Bänziger et al. [9], is an online test administered via the Swiss Centre for Affective Sciences webpage. The task consists of 30 video clips of actors (three for each of 10 emotions: irritation, anger, anxiety, fear, happiness, elated joy, disgust, contempt, sadness, despair) that are presented in four modalities (still picture, video only, audio only, audio with video), yielding a total of 120 items. After viewing each clip, participants are asked to label the emotion shown in the clip. We used the English-language version of the task for our replication.

## 2.5. Affective factors questionnaires

Each participant completed the same affective factor questionnaires used by Palemro et al. [1]. These are Lovibond & Lovibond's [13] DASS, Watson et al.'s [14] PANAS, Baron-Cohen et al.'s [15] AQ, Baron-Cohen & Wheelwright's [16] EQ and Jolliffe & Farrington's [17] BES.

# 3. Procedure

As in Palermo et al. [1], the DASS and PANAS were administered before the emotion-labelling and emotion-matching tasks. The other affective questionnaires were administered after the emotion-labelling and emotion-matching tasks. Following Palermo et al. [1], we ran the emotion-matching task before the emotion-labelling task and ran both of these tasks before the multimodal emotion recognition test.

**Table 1.** Descriptive statistics for all measures after truncation and exclusions ($N = 156$).

| measure | minimum | maximum | mean | s.d. |
|---|---|---|---|---|
| Age | 18 | 37 | 22.87 | 3.74 |
| AQ | 3 | 37 | 20.04 | 6.4 |
| BES affective subscale | 18 | 53 | 39.81 | 7.24 |
| BES cognitive subscale | 24 | 42 | 33.93 | 3.70 |
| DASS anxiety subscale | 0 | 16 | 4.63 | 3.5 |
| DASS depression subscale | 0 | 17 | 4.48 | 4.06 |
| EQ | 17 | 70 | 41.13 | 11.48 |
| PANAS negative affect | 10 | 40 | 20.79 | 6.5 |
| PANAS positive affect | 13 | 49 | 30.66 | 7.75 |
| emotion labelling | 45 | 92.36 | 75.39 | 8.57 |
| emotion matching | 25 | 82.64 | 64.06 | 11.26 |
| MERT | 24 | 74.36 | 53.16 | 9.01 |

All subscale and component scores tested by Palermo et al. [1] were calculated following the instructions for these questionnaires. The scales we used were the anxiety scale of the DASS, the depression scale of the DASS, the positive affect scale of the PANAS, the negative affect scale of the PANAS, the AQ, the EQ scores, scores on the affective component of the BES, and scores on the cognitive component of the BES. These are the same scales employed by Palermo et al. [1]. Cronbach's alpha was high for each scale ($> 0.75$), except the BES cognitive subscale, for which it was 0.62 (see electronic supplementary material for all individual Cronbach's alphas).

## 3.1. Data exclusions and data quality checks

Outliers (scores on a measure that were more than three standard deviations from the mean score for that measure) were adjusted to score one point higher than the closest non-outlier score (following [1]). As a positive control, participants scoring lower than chance on any of the expression recognition tasks were excluded from all analyses. No other exclusions or data manipulations were carried out. Three participants were removed who did not complete one of the emotion tasks and nine participants were removed who had missing questionnaire data (i.e. the final dataset consisted of 156 participants). A total of 10 scores were truncated. Descriptive statistics for all measures are given in table 1. Although we had aimed for a similar ratio of male and female participants to Palermo et al. [1] (55% women, 45% men), the impending Covid-19 crisis and associated lockdown meant that this was not possible. Our final dataset consisted of 113 women, 37 men, and six participants who did not report their sex. Participants who did not report their sex were not included in analyses controlling for participant sex.

# 4. Results

All analysis code, data and full results are publicly available at https://osf.io/kexhr/ and in our electronic supplementary material.

*Hypothesis 1.* Scores on the anxiety scale of the DASS will be significantly negatively correlated with performance on the emotion-matching task.

As in Palermo et al. [1], we tested Hypothesis 1 by calculating Pearson's product-moment correlation coefficient for the relationship between scores on the anxiety scale of the DASS and performance on the emotion-matching task. Our sample size had 80% power to detect effects as small as $|r| = 0.219$ at the 5% significance level.

Scores on the emotion matching task and the DASS anxiety subscale were not significantly correlated ($r = -0.117$, 95% CI $= [-0.269, 0.041]$, $p = 0.147$).

*Hypothesis 2.* Scores on the anxiety scale of the DASS will be significantly negatively correlated with performance on the emotion-labelling task.

**Table 2.** Correlations between affective measures and performance on the emotion recognition tasks. The table shows *r* values, with *p*-values in parentheses. $N = 156$.

| | emotion matching | emotion labelling |
|---|---|---|
| AQ | −0.061 (0.447) | −0.108 (0.181) |
| BES affective subscale | 0.122 (0.130) | 0.058 (0.473) |
| BES cognitive subscale | 0.103 (0.201) | 0.153 (0.057) |
| DASS depression subscale | 0.018 (0.826) | −0.035 (0.665) |
| EQ | 0.085 (0.29) | 0.13 (0.106) |
| PANAS negative affect | −0.088 (0.273) | −0.149 (0.063) |
| PANAS positive affect | −0.094 (0.242) | −0.055 (0.494) |

As in Palermo *et al*. [1], we tested Hypothesis 2 by calculating Pearson's product-moment correlation coefficient for the relationship between scores on the anxiety scale of the DASS and performance on the emotion-labelling task. Our sample size had 80% power to detect effects as small as $|r| = 0.219$ at the 5% significance level.

Scores on the emotion labelling task and the DASS anxiety subscale were significantly negatively correlated ($r = −0.175$, 95% CI = [−0.323, −0.018], $p = 0.029$).

**Hypothesis 3.** Performance on neither the emotion-matching nor emotion-labelling tasks will be significantly correlated with scores on the depression scale of the DASS, the positive affect scale of the PANAS, the negative affect scale of the PANAS, AQ scores, EQ scores, scores on the affective component of the BES, or scores on the cognitive component of the BES.

As in Palermo *et al*. [1], we tested Hypothesis 3 by calculating the Pearson's product-moment correlation coefficients for the relationships between the emotion-matching and emotion-labelling tasks and scores on the depression scale of the DASS, the positive affect scale of the PANAS, the negative affect scale of the PANAS, AQ scores, EQ scores, scores on the affective component of the BES, and scores on the cognitive component of the BES. We used Steiger's test for comparing elements of a correlation matrix [18]. Our sample size had 80% power to detect effects as small as $|r| = 0.219$ at the 5% significance level. These results are summarized in table 2. None of the affective measures were correlated significantly with performance on the emotion matching or labelling tasks.

**Hypothesis 4.** Performance on Bänziger *et al*.'s [9] multimodal emotion recognition test will not be significantly correlated with scores on any of the affective factors.

As in Palermo *et al*. [1], we tested Hypothesis 4 by calculating the Pearson's product-moment correlation coefficients for the relationships between the multimodal emotion recognition test and scores on the depression scale of the DASS, the positive affect scale of the PANAS, the negative affect scale of the PANAS, AQ scores, EQ scores, scores on the affective component of the BES, and scores on the cognitive component of the BES. Our sample size had 80% power to detect effects as small as $|r| = 0.219$ at the 5% significance level. Analysis code for Hypothesis 4 is publicly available at https://osf.io/kexhr/ and in our electronic supplementary material.

These results are summarized in table 3. Only the AQ (negatively), EQ (positively) and BES cognitive subscale (positively) significantly predicted performance on the MERT.

## 4.1. Robustness checks

As in Palermo *et al*. [1], we repeated each analysis using partial correlations to control for possible effects of participant sex, participant age, and both participant sex and participant age simultaneously.

Controlling for sex and age did not alter the pattern of results for the DASS anxiety subscale. Scores on the DASS anxiety subscale negatively and significantly predicted performance on the emotion-labelling task in all of these analyses. By contrast, scores on the DASS anxiety subscale did not significantly predict performance on either the emotion-matching task or the MERT in any analyses. Note that a larger proportion of our participants were women than we had planned in our Stage 1 submission. Because of the large proportion of women in our sample, the results of robustness checks controlling for participant sex should be treated cautiously.

Following Palermo *et al*. [1], we also tested the correlation between each of the affective factors and the first component produced by principal component analysis of scores on the three emotion recognition

**Table 3.** Correlations between affective factors and performance on the MERT. $N = 156$.

|  | R | p-value |
|---|---|---|
| AQ | −0.195 | 0.015 |
| BES affective subscale | 0.155 | 0.153 |
| BES cognitive subscale | 0.237 | 0.003 |
| DASS anxiety subscale | −0.112 | 0.163 |
| DASS depression subscale | 0.013 | 0.868 |
| EQ | 0.238 | 0.003 |
| PANAS negative affect | −0.02 | 0.808 |
| PANAS positive affect | −0.024 | 0.765 |

**Table 4.** Correlations between affective factors and the component produced from a principal component analysis of scores on the three emotion recognition tasks. $N = 156$.

|  | R | p-value |
|---|---|---|
| AQ | −0.146 | 0.069 |
| BES affective subscale | 0.118 | 0.143 |
| BES cognitive subscale | 0.199 | 0.013 |
| DASS anxiety subscale | −0.167 | 0.038 |
| DASS depression subscale | −0.003 | 0.968 |
| EQ | 0.182 | 0.023 |
| PANAS negative affect | −0.109 | 0.177 |
| PANAS positive affect | −0.071 | 0.377 |

tasks. Scores on this component were significantly correlated with anxiety, but no other affective factors, in Palermo et al. [1].

These results are summarized in table 4. The component produced from a principal component analysis of scores on the three emotion recognition tasks was significantly predicted by scores on the BES cognitive subscale, DASS anxiety subscale, and EQ only.

We also conducted additional robustness checks restricting the sample to only those participants ($N = 106$) who scored within the maximum and minimum values for each measure as reported in table 1 of Palermo et al. [1].

In this smaller dataset, scores on the DASS anxiety subscale were negatively and significantly correlated with performance on the emotion-labelling task ($r = -0.245$, $p = 0.011$). Scores on the DASS anxiety subscale were negatively correlated with performance on the emotion-matching task ($r = -0.161$, $p = 0.098$) and the MERT ($r = -0.148$, $p = 0.131$), but these correlations were not significant.

## 4.2. Exploratory analyses of social anxiety

Although data on social anxiety specifically were not collected by Palermo et al. [1], some researchers have suggested that because of fears concerning negative evaluation, social anxiety may be a key correlate of individual differences in emotion recognition (e.g. [19,20]). Consequently, we repeated the analyses described in Hypotheses 1, 2 and 4 (and the related robustness checks) using scores on the brief fear of negative evaluation scale (BFNE; [21]) and the 6-item versions of the social interaction anxiety scale (SIAS) and social phobia scale (SPS) developed by [22]. So as to not interfere with our replication of Palermo et al.'s [1] study, these questionnaires were administered in a fully randomized order at the very end of the study.

Analysis code for these exploratory analyses and full results are publicly available at https://osf.io/kexhr/ and in our supplemental materials. These analyses showed little evidence that the BFNE, SIAS or SPS consistently predicted emotion recognition.

## 4.3. Open data statement

All data and analysis code are publicly available on the Open Science Framework (https://osf.io/kexhr/).

# 5. Discussion

Palermo et al. [1] reported that performance on a range of emotion-recognition tasks, including the first component produced by principal component analysis of scores on these emotion-recognition tasks, was negatively correlated with scores on the DASS anxiety subscale, but not measures of other aspects of affective factors. Here, we replicated Palermo et al.'s [1] study with a larger sample. We carried out this replication because results for individual differences in emotion recognition have often not replicated well.

Replicating Palermo et al. [1], participants who scored higher on the DASS anxiety subscale showed significantly poorer performance on the emotion-labelling task across all analyses. We also replicated Palermo et al.'s finding that participants who scored higher on the DASS anxiety subscale showed significantly poorer emotion recognition as measured by a component produced by principal component analysis of scores on all emotion-recognition tasks. Although participants who scored higher on the DASS anxiety subscale tended to perform more poorly on the emotion-matching task, these correlations were not significant in our study (by contrast with Palermo et al's significant results). Nonetheless, we suggest that, collectively, our results show clear support for Palermo et al.'s claim that the DASS anxiety subscale predicts individual differences in general emotion recognition.

While Palermo et al. [1] observed no significant correlations between scores on any of the other affective factor scales and measures of emotion recognition, we saw some evidence that other affective factors may reliably predict individual differences in emotion recognition. For example, scores on the EQ were positively and significantly correlated with performance on the MERT both in our full sample and in the subsample of participants whose scores fell within the range of scores reported by Palermo et al. [1]. Scores on the EQ were also positively and significantly correlated with performance on the emotion-matching task in the smaller dataset. Participants who scored higher on the EQ also performed significantly better on emotion recognition as measured by a principal component analysis of scores on all three emotion-recognition tasks. While Palermo et al's results suggested that individual differences in emotion recognition were predicted specifically (i.e. uniquely) by scores on the DASS anxiety subscale, our results (and our results for EQ in particular) do not support this claim. However, our results for EQ and emotion recognition are consistent with those of some previous studies that also found that people who scored higher on measures of empathy or lower on measures of autistic-like traits performed better on emotion-recognition tasks (e.g. [5,6]). Indeed, other measures of empathy (i.e. the BES cognitive subscale) and autistic-like traits (i.e. the AQ) also appeared to predict individual differences in emotion recognition (see tables 3 and 4). Nonetheless, we stress here that our results for EQ and emotion recognition be treated cautiously until direct replications have been carried out.

In exploratory analyses, we investigated the possible role in emotion recognition of three measures of social anxiety specifically that were not considered in Palermo et al. By contrast with our results for the DASS anxiety subscale, we saw little evidence that any of these measures significantly predicted emotion recognition. We tentatively propose that these null results for social anxiety suggest that the power of DASS anxiety in predicting emotion recognition is unlikely to reflect individual differences in social anxiety specifically, despite some researchers having previously suggested social anxiety may be particularly important for emotion recognition (e.g. [18,19]). However, it should be noted that both our study and Palermo et al. [1] look at predictors of emotional processing averaged across emotional expressions. We cannot therefore rule out that depression and/or anxiety are associated with differences in the processing of specific emotions.

In conclusion, we found that general emotion recognition was negatively correlated with scores on the DASS anxiety subscale, replicating Palermo et al.'s [1] results. However, by contrast with Palermo et al.'s results, we found that other affective factors, most notably those related to empathy (e.g. the EQ) also appeared to predict general emotion recognition. Collectively, these results support the proposal that affective factors predict individual differences in emotion recognition, but that these correlations are not necessarily specific to measures of general anxiety, such as the DASS anxiety subscale, and may also extend to measures of empathy.

Ethics. All aspects of this project were approved by the University of Glasgow College of Science and Engineering Ethics Committee (application no. 300180047).

Data accessibility. This article has no additional data.

Competing interests. The authors declare no competing interests.

Funding. We received no funding for this study.

Acknowledgement. We thank Jeffrey Girard for helpful feedback on an earlier draft.

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
