## [Reviewer comments · Royal Society Open Science]

Review History

RSOS-182032.R0 (Original submission)

Review form: Reviewer 1 (Alex Jones)

Do you have any ethical concerns with this paper?

No

Have you any concerns about statistical analyses in this paper?

No

Recommendation?

Accept with minor revision

Comments to the Author(s)

This review by Dr Alex Jones, Swansea University.

This registered replication report proposes to replicate the study conducted by Palermo et al (2018), in which individuals with lower levels of anxiety have higher scores on facial expression recognition tasks, but measures of AQ, empathy, affect, or depression do not.

The proposed work is clearly outlined and well justified, and I commend the authors on their open and completed analysis plan, as well as the acquisition of the original study materials to aid the replication attempt. The reasons for attempting the replication are clearly justified, and the proposed sample has sufficient power to detect effects somewhat smaller than the original effects found.

There were only two things that came to mind, one minor, and one that is more related to the inferences made from the existing data and the proposed study.

The first was that it was evident that, for the emotion-labelling task, the authors propose to modify the presentation time from 1000ms to 400ms, and the number of facial expressions available in each category. I wondered if the justification for this might be made explicit, as it is not readily apparent, especially when the authors have obtained the original study materials.

The second was on the analyses in the paper. The authors rightly should reproduce the analyses carried out by Palermo et al. However, I do question whether running a series of correlations is a valid approach for inferring what important features may contribute to individual differences in emotional expression, especially with eight predictors. Would the authors consider an extension of their analysis to include some kind of multiple regression that would be more informative? With 160 participants and 8 predictors, the design is sufficiently powered to detect a relatively small R^2 (~.10). Of course, this is not a barrier to acceptance as the authors have set out to replicate this study, but I think this extension would generally improve the interpretation of the data beyond the original.

Review form: Reviewer 2 (Guillermo Recio)

Do you have any ethical concerns with this paper?

No

Have you any concerns about statistical analyses in this paper?

Yes

Recommendation?

Major revision

Comments to the Author(s)

The authors aim at replicating some of the observed relationships between emotion recognition ability and measures of anxiety, depression, autism (among others) reported by Palermo et al (2018) using a similar design and the same stimuli material in a different sample. The motivation and spirit of the study are clear and interesting, however a number of concerns should be addressed.

1) On page 3, second paragraph, the authors argue that mixed results regarding the relationship between affective factors and individual differences in emotion recognition ability (ERA) can be due to the different methods used to measure ERA across studies. They aim at a direct replication to overcome this issue. However, the authors seem to introduce few changes relative to the original study from Palermo et al. (2018). For example, the original study investigated not only affective but also perceptual factors, measuring individual differences in a perceptual task, but the authors skip the perceptual task in their replication. Also, Palermo et al. (2018) used three measures for ERA, and a factor score built on the three measures. I think this a strength of Palermo et al.'s (2018) study. Such factor scores suffer less from the measurement error or methodological factors associated with single task. Unfortunately the authors omit the

Multimodal Emotion Recognition Test (MERT, Bänziger, Grandjean, & Scherer, 2009) and the factor score resulting from merging different measures. The logic behind all these changes is unclear, and should be justified since some of these changes relative to the original study contradict the claim in page 3 that replication studies have the advantage of using the same measure of ERA. Any discrepancies in their results relative to Palermo et al.'s study could be attributed to those changes.

2) Another concern is that the two measures of individual differences in ERA include hits of two rather easy tasks (emotion-matching and emotion-labelling), in which reaction times are more informative of individual differences in performance than hits (e.g., Wilhelm et al. 2014). This might result in constrained variance for hits, if most subjects approach ceiling. How would the authors make sure they have enough variance in hits? And how would they proceed if they have ceiling effects? Planning an a priori check for variance and having an alternative analysis plan would be important. Although this might exceed the scope of a replication, it would be interesting to extend the findings of the replication by including also measures of RTs. It would be easy to implement and provide better understanding of the research question and address a potential limitation in Palermo et al. (2018).

3) Some information regarding the design was somewhat confusing. The authors aim at replicating the study by Palermo et al. 2018, but then in page 6 they refer to another study by Palermo et al. 2013, and mention two changes introduced in the emotion-labeling task, regarding stimulus presentation times and number of trials presented in each emotion category, but those changes had been already introduced and mentioned by Palermo et al. (2018), bringing therefore confusion. I recommend providing all methodological aspects of the design as they are planned, and then, clearly point out any possible differences with the target study for the replication, but not other studies from the same group. The descriptions of the methods should be as detailed as possible in a Stage 1 pre-registered report. Those descriptions should include for example, size of the stimuli, database, color or B&W?, number of trials, number of stimuli by each gender, detailed trial schema, description of lab equipment, monitor sizes, software to be used, etc.

4) The authors aim at a sample size two and a half times larger than the sample of the original study based on recommendations by Simonsohn (2015). Although the resulting sample of N=160 seems reasonable, it would be more convincing if the authors provided a sample size estimation based on power analyses using effect size estimates from previous studies.

5) A detailed description of the aimed sample, including inclusion and exclusion criteria (not only age and gender) would be much appreciated, because potential participants belonging to clinical population might confound the results.

6) I recommend providing the reliability of the measures in the study and address the issue if the observed correlations exceed the value of the square root of the reliability (see Vul et al. 2009).

Minor points

7) Please provide a brief description of the conditions for a direct replication, e.g. in page 3 after line 22.

8) The title and abstract mention that "evidence that affective factors PREDICT individual differences in ERA is mixed" but the analyses plan correlations, not assuming a specific direction in the relation between variables. Hence "ARE RELATED TO" or "RELATIONSHIP BETWEEN" seems more appropriate for that particular sentence.

9) It is arguable that the two selected tasks from Palermo et al. (2018) is a "particularly comprehensive" measure of ERA, compared with other studies that used up to 16 tasks (e.g. Wilhelm et al. 2014). Besides the authors omitted one measure, i.e. MERT (see my point 1). Please rephrase.

10) Simons 2014 does not appear in the reference list.

References:

Bänziger, T., Grandjean, D., & Scherer, K. R. (2009). Emotion recognition from expressions in face, voice, and body: the Multimodal Emotion Recognition Test (MERT). *Emotion*, 9(5), 691-704.

Vul, Harris, Winkielman & Pashler (2009) Puzzlingly high correlations in fMRI studies of emotion, personality, and social cognition. *Perspectives on Psychological Science*, 4 (3), 274-290.

Decision letter (RSOS-182032.R0)

28-Jan-2019

Dear Dr Jones,

The Editors assigned to your Stage 1 Replication submission ("Do affective factors predict individual differences in facial expression recognition?") have now received comments from reviewers. We would like you to revise your paper in accordance with the referee and editors suggestions which can be found below (not including confidential reports to the Editor). Please note this decision does not guarantee eventual acceptance.

Please submit a copy of your revised paper within three weeks (i.e. by the The author due date is unavailable). If deemed necessary by the Editors, your manuscript will be sent back to one or more of the original reviewers for assessment. If the original reviewers are not available we may invite new reviewers.

When submitting your revised manuscript, you must respond to the comments made by the referees and upload a file "Response to Referees" in the "File Upload" step. Please use this to document how you have responded to the comments, and the adjustments you have made. In order to expedite the processing of the revised manuscript, please be as specific as possible in your response.

Once again, thank you for submitting your manuscript to Royal Society Open Science and I look forward to receiving your revision. If you have any questions at all, please do not hesitate to get in touch. Full author guidelines may be found at <http://rsos.royalsocietypublishing.org/page/replication-studies#AuthorsGuidance>.

Kind regards,
Andrew Dunn
Senior Publishing Editor
Royal Society Open Science
openscience@royalsociety.org

on behalf of Professor Chris Chambers (Registered Reports Editor, Royal Society Open Science)
openscience@royalsociety.org

Associate Editor Comments to Author (Professor Chris Chambers):

Two expert reviewers have now appraised the manuscript. Both find merit in the submission but raise concerns about deviations from the original methodology and the need for additional

methodological detail and justification of design elements. Concerning the suggestion of Reviewer 1 to extend the originally reported analysis -- this is permissible provided the original analysis is also reported. A Major Revision is recommended.

Comments to Author:

Reviewer: 1

Comments to the Author(s)

This review by Dr Alex Jones, Swansea University.

This registered replication report proposes to replicate the study conducted by Palermo et al (2018), in which individuals with lower levels of anxiety have higher scores on facial expression recognition tasks, but measures of AQ, empathy, affect, or depression do not.

The proposed work is clearly outlined and well justified, and I commend the authors on their open and completed analysis plan, as well as the acquisition of the original study materials to aid the replication attempt. The reasons for attempting the replication are clearly justified, and the proposed sample has sufficient power to detect effects somewhat smaller than the original effects found.

There were only two things that came to mind, one minor, and one that is more related to the inferences made from the existing data and the proposed study.

The first was that it was evident that, for the emotion-labelling task, the authors propose to modify the presentation time from 1000ms to 400ms, and the number of facial expressions available in each category. I wondered if the justification for this might be made explicit, as it is not readily apparent, especially when the authors have obtained the original study materials.

The second was on the analyses in the paper. The authors rightly should reproduce the analyses carried out by Palermo et al. However, I do question whether running a series of correlations is a valid approach for inferring what important features may contribute to individual differences in emotional expression, especially with eight predictors. Would the authors consider an extension of their analysis to include some kind of multiple regression that would be more informative? With 160 participants and 8 predictors, the design is sufficiently powered to detect a relatively small R^2 (~.10). Of course, this is not a barrier to acceptance as the authors have set out to replicate this study, but I think this extension would generally improve the interpretation of the data beyond the original.

Reviewer: 2

Comments to the Author(s)

The authors aim at replicating some of the observed relationships between emotion recognition ability and measures of anxiety, depression, autism (among others) reported by Palermo et al (2018) using a similar design and the same stimuli material in a different sample. The motivation and spirit of the study are clear and interesting, however a number of concerns should be addressed.

1) On page 3, second paragraph, the authors argue that mixed results regarding the relationship between affective factors and individual differences in emotion recognition ability (ERA) can be due to the different methods used to measure ERA across studies. They aim at a direct replication to overcome this issue. However, the authors seem to introduce few changes relative to the original study from Palermo et al. (2018). For example, the original study investigated not only affective but also perceptual factors, measuring individual differences in a perceptual task, but the authors skip the perceptual task in their replication. Also, Palermo et al. (2018) used three measures for ERA, and a factor score built on the three measures. I think this a strength of Palermo et al.'s (2018) study. Such factor scores suffer less from the measurement error or

methodological factors associated with single task. Unfortunately the authors omit the Multimodal Emotion Recognition Test (MERT, Bänziger, Grandjean, & Scherer, 2009) and the factor score resulting from merging different measures. The logic behind all these changes is unclear, and should be justified since some of these changes relative to the original study contradict the claim in page 3 that replication studies have the advantage of using the same measure of ERA. Any discrepancies in their results relative to Palermo et al.'s study could be attributed to those changes.

2) Another concern is that the two measures of individual differences in ERA include hits of two rather easy tasks (emotion-matching and emotion-labelling), in which reaction times are more informative of individual differences in performance than hits (e.g., Wilhelm et al. 2014). This might result in constrained variance for hits, if most subjects approach ceiling. How would the authors make sure they have enough variance in hits? And how would they proceed if they have ceiling effects? Planning an a priori check for variance and having an alternative analysis plan would be important. Although this might exceed the scope of a replication, it would be interesting to extend the findings of the replication by including also measures of RTs. It would be easy to implement and provide better understanding of the research question and address a potential limitation in Palermo et al. (2018).

3) Some information regarding the design was somewhat confusing. The authors aim at replicating the study by Palermo et al. 2018, but then in page 6 they refer to another study by Palermo et al. 2013, and mention two changes introduced in the emotion-labeling task, regarding stimulus presentation times and number of trials presented in each emotion category, but those changes had been already introduced and mentioned by Palermo et al. (2018), bringing therefore confusion. I recommend providing all methodological aspects of the design as they are planned, and then, clearly point out any possible differences with the target study for the replication, but not other studies from the same group. The descriptions of the methods should be as detailed as possible in a Stage 1 pre-registered report. Those descriptions should include for example, size of the stimuli, database, color or B&W?, number of trials, number of stimuli by each gender, detailed trial schema, description of lab equipment, monitor sizes, software to be used, etc.

4) The authors aim at a sample size two and a half times larger than the sample of the original study based on recommendations by Simonsohn (2015). Although the resulting sample of N=160 seems reasonable, it would be more convincing if the authors provided a sample size estimation based on power analyses using effect size estimates from previous studies.

5) A detailed description of the aimed sample, including inclusion and exclusion criteria (not only age and gender) would be much appreciated, because potential participants belonging to clinical population might confound the results.

6) I recommend providing the reliability of the measures in the study and address the issue if the observed correlations exceed the value of the square root of the reliability (see Vul et al. 2009).

Minor points

7) Please provide a brief description of the conditions for a direct replication, e.g. in page 3 after line 22.

8) The title and abstract mention that "evidence that affective factors PREDICT individual differences in ERA is mixed" but the analyses plan correlations, not assuming a specific direction in the relation between variables. Hence "ARE RELATED TO" or "RELATIONSHIP BETWEEN" seems more appropriate for that particular sentence.

9) It is arguable that the two selected tasks from Palermo et al. (2018) is a "particularly comprehensive" measure of ERA, compared with other studies that used up to 16 tasks (e.g. Wilhelm et al. 2014). Besides the authors omitted one measure, i.e. MERT (see my point 1). Please rephrase.

10) Simons 2014 does not appear in the reference list.

References:

Bänziger, T., Grandjean, D., & Scherer, K. R. (2009). Emotion recognition from expressions in face, voice, and body: the Multimodal Emotion Recognition Test (MERT). *Emotion*, 9(5), 691-704.

Vul, Harris, Winkielman & Pashler (2009) Puzzlingly high correlations in fMRI studies of emotion, personality, and social cognition. *Perspectives on Psychological Science*, 4 (3), 274-290.

Author's Response to Decision Letter for (RSOS-182032.R0)

See Appendix A.

RSOS-182032.R1 (Revision)**Review form: Reviewer 1 (Alex Jones)**

Do you have any ethical concerns with this paper?

No

Have you any concerns about statistical analyses in this paper?

No

Recommendation?

Accept in principle

Comments to the Author(s)

Thank you for clarifying the points during review. This is a well done piece of work - all the best with the data collection!

Review form: Reviewer 2

Do you have any ethical concerns with this paper?

No

Have you any concerns about statistical analyses in this paper?

No

Recommendation?

Accept with minor revision

Comments to the Author(s)

1) I think there has been a misunderstanding regarding my former point 2. I did not suggest to omit hits analyses (if so, the whole replication would be pointless). My suggestion was "it would be interesting to extend the findings of the replication by including also measures of RTs". Thus, I meant reporting RTs in addition to hits, or at least making the RT data also available for other researchers.

2) Regarding my former point 5, because the reliability of the planned measures is reported for non-clinical populations (see Palermo et al. 2018), I have to insist that it would be important to target only non-clinical population during participants' recruitment.

Decision letter (RSOS-182032.R1)

01-Apr-2019

Dear Dr Jones

On behalf of the Editors, I am pleased to inform you that your Stage 1 Manuscript RSOS-182032.R1 entitled "Are affective factors related to individual differences in facial expression recognition?" deemed suitable for in-principle acceptance in Royal Society Open Science subject to minor revision in accordance with the referee and editor suggestions. Please find their comments at the end of this email.

The reviewers and handling editors have recommended publication, but also suggest some minor revisions to your manuscript. Therefore, I invite you to respond to the comments and revise your manuscript.

Please submit the revised version of your manuscript within 7 days (i.e. by the 09-Apr-2019). If you do not think you will be able to meet this date please let me know immediately.

Full author guidelines can be found here
<http://rsos.royalsocietypublishing.org/page/replication-studies#AuthorsGuidance>

Kind regards
Professor Chris Chambers
Royal Society Open Science
openscience@royalsociety.org

Associate Editor Comments to Author (Professor Chris Chambers):

The reviewers have responded positively to the revised submission, with one reviewer recommending IPA and the other minor revisions. Please attend to the comments of Reviewer 2, either through revision or rebuttal. Provided the response is thorough, IPA should be forthcoming without requiring further in-depth Stage 1 review.

Reviewer comments to Author:

Reviewer: 1

Comments to the Author(s)

Thank you for clarifying the points during review. This is a well done piece of work - all the best with the data collection!

Reviewer: 2

Comments to the Author(s)

1) I think there has been a misunderstanding regarding my former point 2. I did not suggest to omit hits analyses (if so, the whole replication would be pointless). My suggestion was "it would be interesting to extend the findings of the replication by including also measures of RTs". Thus, I meant reporting RTs in addition to hits, or at least making the RT data also available for other researchers.

2) Regarding my former point 5, because the reliability of the planned measures is reported for non-clinical populations (see Palermo et al. 2018), I have to insist that it would important to target only non-clinical population during participants' recruitment.

Author's Response to Decision Letter for (RSOS-182032.R1)

See Appendix B.

Decision letter (RSOS-190699.R0)

26-Apr-2019

Dear Dr Jones

On behalf of the Editor, I am pleased to inform you that your Manuscript RSOS-190699 entitled "Are affective factors related to individual differences in facial expression recognition?" has been accepted in principle for publication in Royal Society Open Science as a Stage 1 Replication.

Please note that you must now register your approved protocol on the Open Science Framework (<https://osf.io/rr>), using the 'Submit your approved Registered Report' option and then the 'Registered Report Protocol Preregistration' option. Please use the Registered Report option even though your article is being accepted as a Stage 1 Replication.

Further into the registration process, in the Journal Title field, enter 'Royal Society Open Science (Replication article type, Results-Blind track)' or 'Royal Society Open Science (Replication article type, Fully Preregistered track)', depending on whether you submitted your Stage 1 manuscript under the Results-Blind track (where the study has already been completed) or the Fully Preregistered track (where the study has yet to be started).

Please note that a time-stamped, independent registration of the protocol is mandatory under journal policy, and manuscripts that do not conform to this requirement cannot be considered at Stage 2. The protocol should be registered unchanged from its current approved state.

If you submitted via the Results-Blind track, please note in the Stage 2 manuscript that the pre-

registration was performed after data analysis (e.g. 'This article received results-blind in-principle acceptance (IPA) at Royal Society Open Science. Following IPA, the accepted Stage 1 version of the manuscript, not including results and discussion, was preregistered on the OSF (URL). This preregistration was performed after data analysis').

If instead you submitted via the Fully Preregistered track then please note in the Stage 2 manuscript that the preregistration was performed before the research was undertaken (e.g. 'This article received in-principle acceptance (IPA) at Royal Society Open Science. Following IPA, the accepted Stage 1 version of the manuscript was preregistered on the OSF (URL). This preregistration was performed prior to the research being undertaken').

Following completion of your study, we invite you to resubmit your paper for peer review as a Stage 2 Replication. Please note that your manuscript can still be rejected for publication at Stage 2 if the Editors consider any of the following conditions to be met:

- The Introduction and methods deviated from the approved Stage 1 submission (required).
- The authors' conclusions were not considered justified given the data.

We encourage you to read the complete guidelines for authors concerning Stage 2 submissions at : <http://rsos.royalsocietypublishing.org/page/replication-studies#AuthorsGuidance>. Please especially note the requirements for data sharing and that withdrawing your manuscript will result in publication of a Withdrawn Registration.

Once again, thank you for submitting your manuscript to Royal Society Open Science and I look forward to receiving your Stage 2 submission. If you have any questions at all, please do not hesitate to get in touch. We look forward to hearing from you shortly with the anticipated submission date for your stage two manuscript.

Kind regards,
Andrew Dunn
Senior Publishing Editor
Royal Society Open Science
openscience@royalsociety.org

on behalf of Professor Chris Chambers (Registered Reports Editor, Royal Society Open Science)
openscience@royalsociety.org

Author's Response to Decision Letter for (RSOS-190699.R0)

See Appendix C.

RSOS-190699.R1 (Revision)

Review form: Reviewer 1 (Alex Jones)

Is the manuscript scientifically sound in its present form?

Yes

Is the language acceptable?

Yes

Do you have any ethical concerns with this paper?

No

Have you any concerns about statistical analyses in this paper?

No

Recommendation?

Accept as is

Comments to the Author(s)

The authors have adhered to their clearly laid out protocol from their original study, and have reported their findings concisely, made explicit any exploratory analyses, and made their materials open.

In the spirit of registered reports, I'd like to congratulate the authors on a rigorous, well executed, and informative study.

Review form: Reviewer 2 (Guillermo Recio)

Is the manuscript scientifically sound in its present form?

Yes

Is the language acceptable?

Yes

Is it clear how to access all supporting data?

Yes

Do you have any ethical concerns with this paper?

No

Have you any concerns about statistical analyses in this paper?

No

Recommendation?

Accept with minor revision

Comments to the Author(s)

1. The authors refer again in several parts of the manuscript to "prediction" to describe the relations between variables, but the analyses included correlations, not assuming a specific direction in the relation between variables. I recommend using "are related to" or similar instead of "predict".
2. Please acknowledge that the unexpected larger number of women in the final sample limits the interpretability of the planned robustness check for sex, and also, that might be a confounding factor in the observed relationships between the variables in the study.
3. Accuracy for the emotion labelling was very high. Please discuss whether the low variance of accuracy data in this task could limit the interpretation of the results regarding hypothesis 2.
4. Same as in Palermo et al. (2018) please provide the Bartlett's test of sphericity, the Kaiser-Meyer-Olkin measure, and the variance explained by the principal component for the PCA

analysis. Optimally, the authors would report also component loadings for each task included in the PCA.

5. The data is available, but was not clear to me. Mentioning the exact name of the files they refer to when some data or analysis is mentioned in the MS would help to find the correct file in the repository.

Decision letter (RSOS-190699.R1)

Dear Professor Jones

On behalf of the Editor, I am pleased to inform you that your Stage 2 Replication submission RSOS-190699.R1 entitled "Are affective factors related to individual differences in facial expression recognition?" has been accepted for publication in Royal Society Open Science subject to minor revision in accordance with the referee suggestions. Please find the referees' comments at the end of this email.

The reviewers and Subject Editor have recommended publication, but also suggest some minor revisions to your manuscript. Therefore, I invite you to respond to the comments and revise your manuscript.

Please also ensure that all the below editorial sections are included where appropriate (a non-exhaustive example is included in an attachment):

- Ethics statement

- Data accessibility

<http://datadryad.org/submit?journalID=RSOS&manu=RSOS-190699.R1>

- Competing interests

- Authors' contributions

- Acknowledgements

- Funding statement

Because the schedule for publication is very tight, it is a condition of publication that you submit the revised version of your manuscript within 7 days (i.e. by the 14-Jul-2020). If you do not think you will be able to meet this date please let me know immediately.

- 1) A text file of the manuscript (tex, txt, rtf, docx or doc), references, tables (including captions) and figure captions. Do not upload a PDF as your "Main Document".
- 2) A separate electronic file of each figure (EPS or print-quality PDF preferred (either format should be produced directly from original creation package), or original software format)
- 3) Included a 100 word media summary of your paper when requested at submission. Please ensure you have entered correct contact details (email, institution and telephone) in your user account
- 4) Included the raw data to support the claims made in your paper. You can either include your data as electronic supplementary material or upload to a repository and include the relevant DOI within your manuscript
- 5) Included your supplementary files in a format you are happy with (no line numbers, Vancouver referencing, track changes removed etc) as these files will NOT be edited in production

Kind regards,
 Professor Chris Chambers
 Royal Society Open Science
 openscience@royalsociety.org

Associate Editor Comments to Author (Professor Chris Chambers):

Comments to the Author:

The Stage 2 manuscript was returned to the two reviewers who assessed the Stage 1 submission. Both are positive about the final manuscript. Reviewer 2 offers a number of specific suggestions for revision. Please note that the reviewer's points 1 and 4 are not required to implement as they add to (or alter) the presentation or analysis plan approved at Stage 1, and so I will leave revisions on these points to the judgement of the authors. Please do respond thoroughly to the reviewer's points 2, 3 and 5 (with point 5 on clarity of data curation being especially important). Provided the authors are able to respond comprehensively to these points in a revision, Stage 2 acceptance should be forthcoming without requiring further in-depth review.

Reviewers' comments to Author:

Reviewer: 1

Comments to the Author(s)

The authors have adhered to their clearly laid out protocol from their original study, and have reported their findings concisely, made explicit any exploratory analyses, and made their materials open.

In the spirit of registered reports, I'd like to congratulate the authors on a rigorous, well executed, and informative study.

Reviewer: 2

Comments to the Author(s)

1. The authors refer again in several parts of the manuscript to "prediction" to describe the relations between variables, but the analyses included correlations, not assuming a specific direction in the relation between variables. I recommend using "are related to" or similar instead of "predict".
2. Please acknowledge that the unexpected larger number of women in the final sample limits the interpretability of the planned robustness check for sex, and also, that might be a confounding factor in the observed relationships between the variables in the study.
3. Accuracy for the emotion labelling was very high. Please discuss whether the low variance of accuracy data in this task could limit the interpretation of the results regarding hypothesis 2.
4. Same as in Palermo et al. (2018) please provide the Bartlett's test of sphericity, the Kaiser-Meyer-Oklín measure, and the variance explained by the principal component for the PCA analysis. Optimally, the authors would report also component loadings for each task included in the PCA.
5. The data is available, but was not clear to me. Mentioning the exact name of the files they refer to when some data or analysis is mentioned in the MS would help to find the correct file in the repository.

Author's Response to Decision Letter for (RSOS-190699.R1)

See Appendix D.

Decision letter (RSOS-190699.R2)

Dear Professor Jones:

It is a pleasure to accept your manuscript entitled "Are affective factors related to individual differences in facial expression recognition?" in its current form for publication in Royal Society Open Science.

on behalf of Professor Chris Chambers (Subject Editor)
openscience@royalsociety.org

Appendix A

We thank the Editor and reviewers for their thoughtful and constructive comments on our submission. We have revised the manuscript in light of these and believe that doing so has strengthened the planned work considerably.

We have also added an additional author (Katherine Button) and exploratory analyses of social anxiety to increase the contribution of the work. These additional social anxiety questionnaires will be administered at the very end of the study to ensure they do not interfere with our replication of the Palermo et al. work.

Additionally, we have added the third emotion recognition task (the MERT), suggested by the reviewers.

Associate Editor Comments to Author (Professor Chris Chambers): Two expert reviewers have now appraised the manuscript. Both find merit in the submission but raise concerns about deviations from the original methodology and the need for additional methodological detail and justification of design elements. Concerning the suggestion of Reviewer 1 to extend the originally reported analysis -- this is permissible provided the original analysis is also reported. A Major Revision is recommended.

We have retained the original analyses, but note here that this type of alternative / additional analysis is precisely the type of analysis that our open data will allow other researchers to carry out.

Comments to Author: Reviewer 1 (Dr Alex Jones, Swansea University)

This registered replication report proposes to replicate the study conducted by Palermo et al (2018), in which individuals with lower levels of anxiety have higher scores on facial expression recognition tasks, but measures of AQ, empathy, affect, or depression do not.

The proposed work is clearly outlined and well justified, and I commend the authors on their open and completed analysis plan, as well as the acquisition of the original study materials to aid the replication attempt. The reasons for attempting the replication are clearly justified, and the proposed sample has sufficient power to detect effects somewhat smaller than the original effects found.

There were only two things that came to mind, one minor, and one that is more related to the inferences made from the existing data and the proposed study.

The first was that it was evident that, for the emotion-labelling task, the authors propose to modify the presentation time from 1000ms to 400ms, and the number of facial expressions available in each category. I wondered if the justification for this might be made explicit, as it is not readily apparent, especially when the authors have obtained the original study materials.

Both of these changes to the task originally developed by Palermo et al. (2013) were also made in the study we plan to replicate (Palermo et al., 2018). We have clarified this point in our methods.

Text added to Methods: “We will use the same emotion-matching task as Palermo et al. (2018). This is the 100-item emotion-labeling task originally developed by Palermo et al. (2013), but with two modifications (presentation time for each face reduced from 1000ms to 400ms and the number of facial expressions in each emotion category being the same). Both of these modifications to the task described in Palermo et al. (2013) were also made in Palermo et al. (2018).”

The second was on the analyses in the paper. The authors rightly should reproduce the analyses carried out by Palermo et al. However, I do question whether running a series of correlations is a valid approach for inferring what important features may contribute to individual differences in emotional expression, especially with eight predictors. Would the authors consider an extension of their analysis to include some kind of multiple regression that would be more informative? With 160 participants and 8 predictors, the design is sufficiently powered to detect a relatively small R^2 (~.10). Of course, this is not a barrier to acceptance as the authors have set out to replicate this study, but I think this extension would generally improve the interpretation of the data beyond the original.

We have opted to follow the analyses in the original paper. The type of additional or alternative analyses described by Reviewer 1 will be possible for other researchers using our open data.

Comments to Author: Reviewer 2

The authors aim at replicating some of the observed relationships between emotion recognition ability and measures of anxiety, depression, autism (among others) reported by Palermo et al (2018) using a similar design and the same stimuli material in a different sample. The motivation and spirit of the study are clear and interesting, however a number of concerns should be addressed.

1) On page 3, second paragraph, the authors argue that mixed results regarding the relationship between affective factors and individual differences in emotion recognition ability (ERA) can be due to the different methods used to measure ERA across studies. They aim at a direct replication to overcome this issue. However, the authors seem to introduce few changes relative to the original study from Palermo et al. (2018). For example, the original study investigated not only affective but also perceptual factors, measuring individual differences in a perceptual task, but the authors skip the perceptual task in their replication. Also, Palermo et al. (2018) used three measures for ERA, and a factor score built on the three measures. I think this a strength of Palermo et al.'s (2018) study. Such factor scores suffer less from the measurement error or methodological factors associated with single task. Unfortunately the authors omit the Multimodal Emotion Recognition Test

(MERT, Bänziger, Grandjean, & Scherer, 2009) and the factor score resulting from merging different measures. The logic behind all these changes is unclear, and should be justified since some of these changes relative to the original study contradict the claim in page 3 that replication studies have the advantage of using the same measure of ERA. Any discrepancies in their results relative to Palermo et al.'s study could be attributed to those changes.

The Palermo et al. (2018) study tests two distinct components that might relate to individual differences in emotion recognition (affective factors and perceptual factors). We will attempt to replicate the affective factors component only. We believe this is justified since the affective factors were analyzed separately from the perceptual factors (see Palermo et al., 2018 Table 3). Palermo et al. (2018) do not suggest at any point that their conclusions for the affective factors are contingent on the perceptual factors being tested in the same study. Furthermore, when analyzed separately, both of the emotion recognition measures that we detailed in our original submission were significantly correlated with anxiety. By contrast, the MERT was not significantly related to anxiety (or any other affective factors). Thus, we respectfully (but strongly) disagree that any differences in our results could be explained by not including the perceptual factors tasks or by not including the MERT. Nonetheless, we have added the MERT to our study.

2) Another concern is that the two measures of individual differences in ERA include hits of two rather easy tasks (emotion-matching and emotion-labelling), in which reaction times are more informative of individual differences in performance than hits (e.g., Wilhelm et al. 2014). This might result in constrained variance for hits, if most subjects approach ceiling. How would the authors make sure they have enough variance in hits? And how would they proceed if they have ceiling effects? Planning an a priori check for variance and having an alternative analysis plan would be important. Although this might exceed the scope of a replication, it would be interesting to extend the findings of the replication by including also measures of RTs. It would be easy to implement and provide better understanding of the research question and address a potential limitation in Palermo et al. (2018).

Analyzing hits is essential to replicate the analyses reported in Palermo et al. (2018).

We will, however, address this issue by carrying out additional robustness checks repeating the analyses on only those participants who scored within the maximum and minimum values on each measure as reported in Table 1 of Palermo et al. (2018).

Text added to Analysis plan: "We will also conduct additional robustness checks restricting the sample to only those participants who scored within the maximum and minimum values for each measure as reported in Table 1 of Palermo et al. (2018)."

3) Some information regarding the design was somewhat confusing. The authors aim at replicating the study by Palermo et al. 2018, but then in page 6

they refer to another study by Palermo et al. 2013, and mention two changes introduced in the emotion-labeling task, regarding stimulus presentation times and number of trials presented in each emotion category, but those changes had been already introduced and mentioned by Palermo et al. (2018), bringing therefore confusion. I recommend providing all methodological aspects of the design as they are planned, and then, clearly point out any possible differences with the target study for the replication, but not other studies from the same group. The descriptions of the methods should be as detailed as possible in a Stage 1 pre-registered report. Those descriptions should include for example, size of the stimuli, database, color or B&W?, number of trials, number of stimuli by each gender, detailed trial schema, description of lab equipment, monitor sizes, software to be used, etc.

We have clarified that the two changes were also made in Palermo et al. (2018).

Text added to Methods: "We will use the same emotion-matching task as Palermo et al. (2018). This is the 100-item emotion-labeling task originally developed by Palermo et al. (2013), but with two modifications (presentation time for each face reduced from 1000ms to 400ms and the number of facial expressions in each emotion category being the same). Both of these modifications to the task described in Palermo et al. (2013) were also made in Palermo et al. (2018)."

We have also clarified the additional methods details requested by Reviewer 2 (image size, image database, monitor type).

4) The authors aim at a sample size two and a half times larger than the sample of the original study based on recommendations by Simonsohn (2015). Although the resulting sample of N=160 seems reasonable, it would be more convincing if the authors provided a sample size estimation based on power analyses using effect size estimates from previous studies.

We did provide power estimates in our analysis plan.

Text from Analysis plan: "Our sample size has 80% power to detect effects as small as $|r| = .219$ at the 5% significance level." We note here that this is a smaller r than the focal tests reported by Palermo et al. (significant $|r|$ values ranged from .26 to .32).

5) A detailed description of the aimed sample, including inclusion and exclusion criteria (not only age and gender) would be much appreciated, because potential participants belonging to clinical population might confound the results.

We will address this issue by carrying out additional robustness checks repeating the analyses on only those participants who scored within the maximum and minimum values on each measure as reported in Table 1 of Palermo et al. (2018).

Text added to Analysis plan: “We will also conduct additional robustness checks restricting the sample to only those participants who scored within the maximum and minimum values for each measure as reported in Table 1 of Palermo et al. (2018).”

6) I recommend providing the reliability of the measures in the study and address the issue if the observed correlations exceed the value of the square root of the reliability (see Vul et al. 2009).

We will report reliabilities.

Text added to Methods: Although Palermo et al. (2018) do not report the reliability of these measures, we will report Cronbach’s alpha for each scale.

Minor points

7) Please provide a brief description of the conditions for a direct replication, e.g. in page 3 after line 22.

We have clarified that we mean using the same measures.

Text added to Introduction: Direct replications (i.e., studies using the same measures as the original work) are one way to address this difficulty because they allow for more direct comparison of results across studies (Simons, 2014).

8) The title and abstract mention that “evidence that affective factors PREDICT individual differences in ERA is mixed” but the analyses plan correlations, not assuming an specific direction in the relation between variables. Hence “ARE RELATED TO” or “RELATIONSHIP BETWEEN” seems more appropriate for that particular sentence.

We have made these changes

New title: Are affective factors related to individual differences in facial expression recognition?

New sentence in Abstract: Evidence that affective factors (e.g., anxiety, depression, affect) are significantly related to individual differences in emotion recognition is mixed. Palermo et al. (2018 Journal of Experimental Psychology: Human Perception and Performance) recently reported that individuals who scored lower in anxiety performed significantly better on two measures of facial-expression recognition (emotion-matching and emotion-labeling tasks), but not a third measure (the Multimodal Emotion Recognition Test).”

9) It is arguable that the two selected tasks from Palermo et al. (2018) is a “particularly comprehensive” measure of ERA, compared with other studies that used up to 16 tasks (e.g. Wilhelm et al. 2014). Besides the authors omitted one measure, i.e. MERT (see my point 1). Please rephrase.

We have edited this sentence in the Abstract and also the corresponding section of the Introduction.

Sentence in Abstract: Because the range of affective factors considered in this study and its use of multiple expression-recognition tasks mean that it is a relatively comprehensive investigation of the role of affective factors in facial expression recognition, we propose to carry out a direct replication.

Sentence in Introduction: We chose this particular study to replicate because it considered a relatively broad range of affective factors (various measures of anxiety, depression, mood, and empathy) and showed consistent results across two recently developed comprehensive facial expression recognition tasks (the emotion-matching and emotion-labeling tasks developed and described in Palermo et al., 2013).

10) Simons 2014 does not appear in the reference list.

We have added this reference to the list.

Appendix B

1) I think there has been a misunderstanding regarding my former point 2. I did not suggest to omit hits analyses (if so, the whole replication would be pointless). My suggestion was "it would be interesting to extend the findings of the replication by including also measures of RTs". Thus, I meant reporting RTs in addition to hits, or at least making the RT data also available for other researchers.

We are reluctant to include RTs in our analyses, since the study design is not optimized for collecting RT data. For example, participants in the original study were not instructed to respond as quickly and accurately as possible (arguably critical for RT data to be useful) and response-key mapping was not counterbalanced. For this reason we would prefer not to analyze or record RT data.

2) Regarding my former point 5, because the reliability of the planned measures is reported for non-clinical populations (see Palermo et al. 2018), I have to insist that it would be important to target only non-clinical population during participants' recruitment.

Again, the original study did not constrain participants in this way, so this type of constraint on recruitment would represent a deviation from the original study protocol. We note here that we state we will carry out robustness checks where data points outside the range reported in the original study are excluded, which should address concerns about the possibility we have participants outside the range tested in the original study.

Appendix C

Dear Editors,

Here we submit our stage two manuscript.

Best wishes, Ben Jones

Appendix D

Response to reviewers' comments

1. The authors refer again in several parts of the manuscript to "prediction" to describe the relations between variables, but the analyses included correlations, not assuming a specific direction in the relation between variables. I recommend using "are related to" or similar instead of "predict".

We politely disagree with R2 that 'predict' necessarily implies causality, so have retained this terminology.

2. Please acknowledge that the unexpected larger number of women in the final sample limits the interpretability of the planned robustness check for sex, and also, that might be a confounding factor in the observed relationships between the variables in the study.

We now note that the results of robustness checks controlling for participant sex should be treated cautiously because of the large proportion of women in our sample.

Text on pages 13 and 14: "Note that a larger proportion of our participants were women than we had planned in our Stage 1 submission. Because of the large proportion of women in our sample, the results of robustness checks controlling for participant sex should be treated cautiously."

3. Accuracy for the emotion labelling was very high. Please discuss whether the low variance of accuracy data in this task could limit the interpretation of the results regarding hypothesis 2.

We admit to being somewhat puzzled by this suggestion. We replicate the original finding that anxiety predicted performance on the emotion labelling task in all analyses, suggesting that a lack of variability in scores on this task is not preventing us from observing significant correlations. Indeed, we also see a significant correlation between anxiety and scores on the labelling task on a subset of the sample that falls within the range of scores reported in the original study. Additionally, we replicate the finding from the original study that scores on all three recognition measures load strongly on the first component produced by PCA of scores on the three recognition tasks. Given these points, we think it would be misleading to suggest that the level of accuracy we see on the labelling task is problematic.

4. Same as in Palermo et al. (2018) please provide the Bartlett's test of sphericity, the Kaiser-Meyer-Olkin measure, and the variance explained by the principal component for the PCA analysis. Optimally, the authors would report also component loadings for each task included in the PCA.

As the editor noted, these additional statistics deviate from those we said we would include in our Stage 1 analysis plan. Consequently, we have not included them. Of course, since the data are available, interested readers can calculate these additional statistics for themselves if they want to.

5. The data is available, but was not clear to me. Mentioning the exact name of the files they refer to when some data or analysis is mentioned in the MS would help to find the correct file in the repository.

We apologise for the confusion. This issue seems to have arisen from our analysis code for the stage one and stage two submissions being unclearly labelled. We have added a readme.txt file to the OSF page clarifying the file names. The readme.txt also notes that the analysis code details all initial data wrangling and coding.